# From Deficit to Strength-Based Aboriginal Health Research—Moving toward Flourishing

**DOI:** 10.3390/ijerph20075395

**Published:** 2023-04-04

**Authors:** Jonathan Bullen, Trish Hill-Wall, Kate Anderson, Alex Brown, Clint Bracknell, Elizabeth A. Newnham, Gail Garvey, Lea Waters

**Affiliations:** 1EnAble Institute, Curtin University, Perth, WA 6102, Australia; 2Telethon Kids Institute, Perth, WA 6009, Australia; alex.brown@telethonkids.org.au; 3Faculty of Medicine, The School of Public Health, The University of Queensland, Herston, QLD 4006, Australia; 4National Centre for Indigenous Genomics, The John Curtin School of Medical Research, Australian National University, Canberra, ACT 2601, Australia; 5School of Languages and Cultures, The University of Queensland, St. Lucia, QLD 4067, Australia; 6School of Population Health, Curtin University, Perth, WA 6102, Australia; 7FXB Center for Health and Human Rights, Harvard University, Boston, MA 02115, USA; 8Centre for Wellbeing Science, Melbourne Graduate School of Education, The University of Melbourne, Parkville, VIC 3101, Australia

**Keywords:** Aboriginal, First Nations, wellbeing, salutogenesis, flourishing, positive psychology, complex systems

## Abstract

Aboriginal Australians have a fundamental human right to opportunities that lead to healthy and flourishing lives. While the impact of trauma on Aboriginal Australians is well-documented, a pervasive deficit narrative that focuses on problems and pathology persists in research and policy discourse. This narrative risks further exacerbating Aboriginal disadvantage through a focus on ‘fixing what is wrong’ with Aboriginal Australians and the internalising of these narratives by Aboriginal Australians. While a growing body of research adopts strength-based models, limited research has sought to explore Aboriginal flourishing. This conceptual paper seeks to contribute to a burgeoning paradigm shift in Aboriginal research, seeking to understand what can be learned from Aboriginal people who flourish, how we best determine this, and in what contexts this can be impactful. Within, we argue the case for a new approach to exploring Aboriginal wellbeing that integrates salutogenic, positive psychology concepts with complex systems theory to understand and promote Aboriginal wellbeing and flourishing. While deeper work may be required to establish the parameters of a strength-based, culturally aligned Aboriginal conceptualisation of positive psychology, we suggest the integration of Aboriginal and Western methodologies offers a unique and potent means of shifting the dial on seemingly intractable problems.

## 1. Introduction

“*Aboriginal Law refers to a complex relationship between humanity and land which extends to cover every aspect of life; to that extent it is what theorists call a ‘complex system’, in that it explains both the observer and the observed.*”Mary Graham—Kombu-Merri and Wakka Wakka person

“*the purpose of models is not to fit the data, but to sharpen the questions.*”Samuel Karlin—mathematician

### 1.1. Positionality

We acknowledge and pay respect to the traditional owners and custodians of the Nyoongar boodja on which the first two authors—mother and son and *Wardandi Nyoongar* people—conceived this manuscript. We wish to acknowledge the continuing connection to land, sea and community and we pay our respects to our Elders past and present. We also acknowledge all Aboriginal peoples across this continent now known as Australia.

This paper emerged through considerable discussion between the first two authors, a Wardandi Noongar (the Noongar are the Indigenous people of Western Australia’s south-west) mother and her eldest son, about Aboriginal health and wellbeing and the possible factors across each of their lives that has led to their own current states of wellbeing. Both have endured despite unique and distinct experiences of the Stolen Generation and its impacts: one taken from her family as a child never to know her biological parents, the other of the first generation in his family to be raised by his biological parents since the Aborigines Act was passed in Western Australia in 1905 [1]. Both belong to an Aboriginal family that, while deeply impacted by the policies and practice of Australia’s colonial legacy, has over time reconceptualised what it means to be ‘doing well’ as Aboriginal people.

As a team of Indigenous and non-Indigenous people and researchers, our approach to this conceptual paper was inherently shaped by principles of Indigenist research methods [2], specifically the privileging and prioritisation of Aboriginal Australians perspectives and ways of knowing, doing and being throughout. This is strongly aligned with the National Aboriginal and Torres Strait Islander Health Plan 2013–2023 overarching principles of ‘Health Equality and a Human Rights Approach’ and ‘Aboriginal and Torres Strait Islander community control and engagement’. Respectively, they stipulate that “the principles of the United Nations Declaration on the Rights of Indigenous Peoples and other human rights instruments support Aboriginal and Torres Strait Islander people in attaining the highest standard of physical and mental and social health,“ and that there must be “…full and ongoing participation by Aboriginal and Torres Strait Islander people and organizations in all levels of decision-making affecting their health needs” [3] (p. 10). We recognise the importance of reflecting on and articulating our positionality in terms of our own backgrounds, perspectives, and values that we each bring to the paper [4,5]. The first author (J.B.) is a Wardandi Noongar man and early career researcher, with an interest in expanding the lens through which Aboriginal health and wellbeing is viewed, understood, and promoted. The second author (T.H.W.) is a Wardandi Noongar woman, Stolen Generations survivor and flourisher, Aboriginal health and wellbeing educator, and the mother of the lead author. The third author (K.A.) is a non-Indigenous Australian senior researcher, living and working on Gubbi Gubbi and Jinibara country, experienced in conducting collaborative qualitative research with Indigenous researchers and communities. The fourth author (A.B.) is a Yuin man and senior researcher with extensive experience in public health services, infectious diseases and chronic disease care, health care policy and research. The fifth author (C.B.) is a Noongar man from Western Australia’s south coast and a mid-career researcher working at the intersection of Indigenous song, language, and landscape. The sixth author (E.N.) is a non-Indigenous woman and mid-career researcher living and working on Noongar country at the nexus of mental health, trauma and adversity. The seventh author (G.G.) is a Kamilaroi woman, descendant of a Stolen Generations survivor, and senior researcher with extensive research experience in Indigenous health and wellbeing. The eighth author (L.W.) is a non-Indigenous woman living and working on Wurundjeri country, senior researcher and expert in positive psychology. The collective voice across the manuscript’s Aboriginal and *Wadjella* (non-Aboriginal) authorship draws on the diverse cultural and disciplinary perspectives to make a case for expanding how we conceptualise Aboriginal health and wellbeing research, and what this may mean for Aboriginal health and wellbeing more broadly.

### 1.2. Background

As enshrined for all individuals and populations, Aboriginal Australians have a fundamental human right to opportunities that lead to healthy and flourishing lives, despite any adversity they face [6]. However, the prevalence and causes of trauma for Aboriginal people [7], and the collective and pervasive impact of this trauma on the wellbeing of Aboriginal Australians is well-documented [8]. The impacts of past policies and practices upon Indigenous people have left an enduring legacy of physical and psychological ill health, intergenerational trauma, loss of identity, family and community disadvantage and dysfunction, and disconnection from country and culture [7,8]. These experiences and impacts parallel those of Indigenous peoples of colonized countries globally [9]. Disturbingly, the impacts experienced by survivors of Australia’s Stolen Generation (the thousands of Indigenous Australian’s forcibly removed from their homes, families and country to be placed in institutions, foster care, or adopted by non-Indigenous families, as a consequence of Australian governmental policies from approximately 1850 through to the 1970’s) and their descendants are considered to be even greater than those witnessed among the broader Aboriginal Australian population [8]. Australian governments have responded with a range of strategic policy, program and resourcing initiatives that seek to address disparities in health outcomes. However, these initiatives have met with limited success [10]. While research has examined the deleterious effects of colonisation and past/on-going policies on Aboriginal people broadly, a pervasive deficit narrative that focuses on problems and pathology dominates academic and policy discourse. This narrative risks further exacerbating Aboriginal disadvantage through the production of evidence that focuses on ‘fixing what is wrong’ with Aboriginal Australians, and perhaps more problematically, the subsequent internalising of deficit narratives of self among Aboriginal Australians [11].

There are notable examples in the broader literature of Aboriginal Australian people who are living, and redefining what it means to be healthy [12]. However, few researchers actively assume a strength-based approach to better understand factors that enable some Aboriginal families and individuals to attain high levels of subjective wellbeing, to ‘flourish’ [13]. This is particularly so for those who are powerfully affected by historical and on-going trauma. Even fewer are the examples within the literature of those factors that contribute to and underpin Aboriginal wellbeing [14,15,16]. This raises an important question about the underlying factors that support the flourishing of some Aboriginal Australians in spite of their experiences of substantial trauma. Put one way, this question might be: ‘Why do some Aboriginal families and people thrive while others do not?’ However, the causes of Aboriginal health and wellbeing disparity and the lived experience of this disparity are well documented and not the focus of this paper. Rather, when conceptualising this paper, we instead framed our thinking with an alternative question: ‘For those Aboriginal families and people who are doing well and who flourish, what needs to be true?’

In doing so, we seek to contribute to a burgeoning paradigm shift in approaches to Aboriginal research that seeks to understand what can be learned from Aboriginal people, families and communities who are doing well. While an important goal in its own right, we extend this by asking how we best determine these things, and in what contexts might we see these learnings take shape and have relevance, influence and impact.

This paper works on the proposition that positive psychology is an intuitive and useful way to bring together a range of disparate constructs/concepts and facilitate their integration, utilisation and interpretation within our research processes. Within this manuscript, we first discuss Aboriginal wellbeing as a multifaceted construct and the complexity of causation, noting the broader system of factors that can contribute to disadvantage. We then argue the case for an approach to exploring Aboriginal wellbeing that integrates salutogenic, positive psychology concepts with complex systems theory to understand and promote Aboriginal wellbeing and flourishing.

We briefly address existing conceptualisations of strength-based Aboriginal research, then elaborate on potential opportunities and tensions within this research approach, before concluding with a consideration of its fit to Aboriginal health and wellbeing and a discussion of ways forward.

## 2. Wellbeing as a Multifaceted Construct

Wellbeing for Aboriginal peoples is a holistic, multifaceted construct [15]. Recent Aboriginal-led research has sought to establish models to understand, interpret and promote what wellbeing means for Aboriginal Australians [14,15,16]. Fundamentally driven by Aboriginal research methodologies [2], this body of work seeks to ensure and facilitate culturally aligned methods and definitions of wellbeing as an empowering and emancipatory process to guide examinations of wellbeing. The adoption of such an approach enables the translation of findings to new, improved and appropriate models of wellbeing to overcome entrenched inequality. These models are noted as interconnected and include a broad range of components considered important to Aboriginal people. These include culture, community and family, but also a sense of belonging, a sense of purpose and control over one’s life, and sufficient economic means [14]. Importantly, this work suggests the culturally bound nature of the features of life important to Aboriginal populations, compared with non-Aboriginal developed models. Related to this, this work also suggests conceptual differences in those factors considered meaningful to wellbeing, but also their weighting relative to other more commonly used metrics. In contrast to this still developing evidence, most research into Aboriginal social and emotional wellbeing has focused on the reduction of pathology, falling well short of intended aims [10]. Similarly prevalent has been its framing in a deficit discourse consisting of representations of Aboriginal people and populations entrenched in a narrative of deficiency, negativity and failure [17]. Despite renewed focus on greater involvement of Aboriginal Australians in determining targets of relevance to Aboriginal communities [18], the focus remains almost solely one of reducing deficit and pathology compared with Western ‘objective’ measures of wellbeing [19]. These contrasting ideas rest on a tension inherent to much of Aboriginal Australian existence since colonisation around who gets to be the arbiter of things for our families and communities. In this manuscript’s context, the question of who defines wellbeing and flourishing, and for whom, is relatively straightforward: Aboriginal people and their communities must determine what each means to them, and while there is likely to be overlap, this definition may differ from community to community. This self-determination of wellbeing and flourishing is deeply important in the Aboriginal Australian context, the outcome likely reflecting differences between Aboriginal holistic concepts of wellbeing and Western reductionist concepts [14,15,16]. To the best of our knowledge, very little literature has explored Indigenous flourishing, with only a few examples in the global Indigenous context, including in Aotearoa [20], Canada [21], Australia [22], and the United States [23]. Each has varied significantly in terms of research methods, and definitions and measurement of flourishing. Each also shares significant similarity in the use of culturally exogenous measures and thus definitions of flourishing (with the exception of [20]). This problematic notion of a universal construct of flourishing has been recently discussed [13].

Accordingly, what has received little focus within Aboriginal Australian health and wellbeing contexts are the factors that develop and enhance health and wellbeing, extending into the realm of a ‘life well lived’—in short, flourishing [24], as defined by Aboriginal Australians. Similarly, little focus has been given to how this is facilitated within the broader system of factors for Aboriginal populations. In the general population, there is a well-established evidence base of strength-based research exploring the lives and experiences of flourishing individuals and families, despite experiences (historical and/or on-going) of trauma [25]. However, little is known about the factors that influence the life trajectory of flourishing Aboriginal families and individuals. While Aboriginal flourishing is significant in its own right, it is doubly so when considered in light of the fact that “*those with positive post-event trajectories of resilience and growth […] are not typically seen by clinicians, nor studied by clinical researchers*” [26]. That many Australian Aboriginal families and individuals have found their way to flourishing is a topic of national significance.

### Complexity in Origin and Solution?

The commencement and maintenance of Aboriginal health trajectories are founded on a complex body of determinants [27,28,29]—for example, socioeconomic status, educational access, housing, transportation, behavioural factors, community capacity and support, and the experience of discrimination. These common distal determinants are linked to, mediate, and are often reinforced by, more proximal determinants such as genetic, health behavioural and socio-environmental interaction factors [30]. Each are deeply embedded in colonised societies. While the existence and impact of these discreet determinants are well understood, less so are the exponential impacts of their interrelatedness [31,32]. These are complex issues difficult to disentangle; their continuing consequences make them deeply challenging to understand and change, which is exemplified by a persistent high burden of disease and inter-generational disadvantage [33,34]. While embedded within our scientific systems, linear approaches to understanding and improving Aboriginal health outcomes often fall short when considered in the context of the ‘wicked’ nature of multiple interrelated factors. Other, more complexity-oriented approaches are useful, perhaps necessary, to disentangle the ‘tangled’ and make clear emergent properties, states, constructs and points of leverage that may prove useful in working with populations that struggle under the weight of the impact of such complexity [35,36].

## 3. A Strength-Based Complex Systems Approach to Understanding Aboriginal Flourishing

We propose a paradigmatic shift framed by systems-informed positive psychology [37]. Essentially, systems-informed positive psychology is based upon two key theoretical propositions. The first, positive psychology [38], focuses on exploring the ‘well-being, contentment and satisfaction (in the past); hope and optimism (for the future); and flow and happiness (in the present)’ [p. 5] of individuals and family. Aligned with calls for greater research focus on the many strengths within Aboriginal communities, positive psychology advocates for a fundamental shift in the way the world is viewed that consists of a distinction and focus inclusive of opportunities for growth and flourishing, as opposed to solely ‘weathering the storm’ [39]. Far from denying adversity, it is via the integration and understanding of positive and negative aspects of life—experiences, behaviours and practices of individuals and families, and their sociocultural and ecological contexts—that the mechanisms that facilitate and promote valued outcomes arise [15]. While positive psychology has generated meaningful outcomes for many populations, there are critiques of both the heterogenous population sampled and impacted, and the context and settings most positive psychology interventions have been conducted within [40]. Only recently has there been an increasing focus upon and relevance to non-WEIRD (White, Educated, Industrialised, Rich and Democratic) populations [41]. At the individual and community level, there is good evidence that these salutogenic approaches can be beneficial [42,43,44]. Studies at the individual level have noted the efficacy of a range of interventions, including the development of character strengths, improving wellbeing, reducing depressive symptoms, and improving work and study performance [42]. At the community level, studies have targeted several areas, including psychological wellbeing, interpersonal wellbeing, occupational wellbeing, and character strength development [44]. Despite a community focus, these studies have been critiqued for essentially being a ‘group version’ of successful individual level positive psychology interventions, and for ignoring context, social justice and values [36,37,44,45]. This conceptualisation of the community as an aggregation of individuals and their wellbeing differs greatly from the notion of a community as a complex organism interacting and intersecting across an array of factors (including the individuals themselves) to generate wellbeing [44]. This is an important point.

The second, systems theory [46], posits that a system is ‘an interconnected set of elements that is coherently organized in a way that achieves something’ [46] (p. 11). Accordingly, systems theory avoids simplistic, granular perspectives of phenomena, instead concerning itself with the ways in which the diverse functions of discreet elements within a bounded system dynamically interact. Certainly, it enables a focus on understanding the state of an element at any given time by examining the interconnectedness between elements within the bounds of the broader system. However, its real strength lies in its capacity to explicate how and why complex phenomena self-organise, coming together in concert to facilitate elemental adaptation, and/or the emergence of complex phenomena previously unknown or unanticipated [47].

Systems-informed positive psychology brings the two theories together through a foundation of epistemological, political and ethical assumptions of the world [37], specifically that an objective reality likely exists, alongside simultaneous and multiple subjective perspectives of that reality. There is constant negotiation, granting and embodiment of rights, responsibilities and power by people within a given system; the notion of wellbeing must be defined and redefined collectively, and thus move toward what is good, right, and optimal for the collective. At present, it is unknown whether these assumptions hold for Aboriginal populations, and thus, some exploration, interrogation and redefinition is required. However, in short, systems-informed positive psychology, in a human-oriented psychosocial and sociocultural context, acknowledges that development of entities toward individual or collective wellbeing does not occur in a linear, isolated way, but recognises the broader forces that play a vital role in the shape of these at any given point in time [37,45].

There is evidence that familial, social, economic, cultural, educational, environmental, and other factors are important to subjective wellbeing within Aboriginal communities [12]. Moreover, it is the balance and interconnectedness of these threads of wellbeing that contribute to holistic wellbeing in Aboriginal society [14,48], aligning with the principles of systems science [37,46,47]. Thus, despite contestation over what fundamentally constitutes both wellbeing and community [49], the notion of community wellbeing as more than the sum of its parts (not simply an aggregation of these parts) resonates strongly with Indigenous cultural concepts of relationality, and frameworks of Indigenous wellbeing [14,15,16]. However, there is little research examining the complexity and interaction between these, including within and between individuals, social networks and systems and the broader sociocultural ecology distinct to Aboriginal Australian communities over the lifespan. Moreover, the influence of these factors that catalyse trajectories toward, and emergence of, a positive psychology of wellbeing is unexplored in the Aboriginal Australian context. Our proposed approach considers key principles of complex systems, acknowledging the temporal, nonlinear nature of human development as individuals, families and communities move through their lives [46,47]. Fundamentally shaped by the concept of salutogenesis, it focuses on and explores the precursors, causes and maintainers of health and wellbeing, as opposed to an exclusive focus on prevention and elimination of disease from a baseline of disease and individual unwellness [50]. This adoption, integration, and understanding of the potential of strength-based systems frameworks within Aboriginal wellbeing research are particularly important when considered in light of holistic Aboriginal conceptualisations of wellbeing and what is going ‘right’ with our communities [12,14,15,16]. Of additional importance is evidence linking these broader conceptualisations of psychosocial wellbeing to important potentialities and possibilities across a broad range of physiological health interventions and outcomes [51], from both practical and theoretical vantage points.

### 3.1. Strength-Based Aboriginal Research: Gathering Momentum

Approaches to research that focus on, promote, and /or incorporate the strengths and knowledge of Australian Aboriginal communities are growing in their usage. Adopted for their capacity to disrupt a history of deficit approaches to research, these approaches push back against the potential harms of deficit narratives to Aboriginal people and communities [11,52]. There is a growing body of research informing policy and practice that might be described as strength-based and that is incorporated within research teams, the governance models and methodology. Still maturing, this corpus of knowledge founded on Aboriginal methodologies [2] increasingly includes and foregrounds concepts of co-design. In short, this is the fundamental inclusion and foregrounding of Aboriginal worldviews and voices within that drive the research process [53] and are intended to set the conditions for appropriate engagement for a given context. The way this is enacted spans the spectrum of quality and authenticity, ranging from on-going tokenistic approaches that remain little more than outdated models of ‘consultation’ [54], to richly comprehensive methodologies guided, informed and owned by Aboriginal people and communities throughout the entire research life cycle (for example, [55]).

Within this, the centrality of culture is also growing in primacy [52] and being recognised as fundamental by governments and funding bodies (we note that Aboriginal people have long understood this). There is a growing body of work spanning cultural practice in recent times, from Aboriginal-determined models and frameworks of social and emotional wellbeing [14,16] to projects focused on language revitalisation [56], giving voice to country through song [57], and birthing on country [58]. Here we briefly expand on a few pieces of research grounded in strengths and their potential fit to a salutogenic model.

Sivak et al.’s [56] research utilised a social and emotional wellbeing framework [16] to explore findings from their work on language revitalisation, specifically highlighting the positive psychological and social and emotional wellbeing impacts of participation. Profound meaning was noted as being derived from several sources, including the immediate process of language revitalisation (in terms of connection and/or re-connection to culture and country), but also in terms of relationships within and between generations and the passing on of this knowledge to descendants. Participants articulated experiences of strong positive emotion as a result of these things, but also from the sense of belonging to, and leading, a community initiative that could bring together and heal wounds between Aboriginal and non-Aboriginal communities. This last point speaks clearly to the importance and impact derived from meaningful engagement in research, as noted elsewhere [59].

McBride et al. [60] explored Aboriginal women’s conceptualisation of cardiovascular health, foregrounding wellbeing from a cultural perspective and highlighting the protective and risk factors recognised by participants, but also the gap between this recognition and the broader health system response. Importantly, their work speaks strongly to the linkages and interactions between culture, the broader system of factors (such as the environment, health services, and policy) and components of psychological wellbeing (such as meaning, relationships, and autonomy) and physiological wellbeing. These latter two points, and their interrelationship, have been noted strongly in the broader literature (for example, [51]).

Very recent research has more explicitly aligned with tenets of positive psychology and wellbeing, albeit across a broad populace sample. Sofija et al.’s [22] exploration of the predictors of flourishing for emerging adults in Australia highlighted greater rates of flourishing in Aboriginal populations relative to non-Aboriginal populations. Findings noted (1) the possibility of a trauma- and disadvantage-derived resilience and/or (2) the existence of inherent strengths specific to this subpopulation that facilitate flourishing. While the former has been well-documented in Aboriginal populations (for example, [61,62,63]), the latter has only very recently received more rigorous attention.

Most of this work has not been conceptualised within a positive psychology theoretical framework. Indeed, to our knowledge, no published research in Aboriginal health and wellbeing has adopted a systems-informed positive psychology approach. However, it is relatively simple to understand and interpret the alignment of these projects with positive psychology concepts and, importantly, a salutogenic rather than pathogenic positioning. Our aim is to make clear the synergies between positive psychology, Aboriginal research and Aboriginal research methodologies. We argue that it is possible to do so without losing those elements vital to Aboriginal wellbeing research, such as the pre-eminence of culture, relationality or of Aboriginal methodology and worldview—these are not mutually exclusive ideas [64]. While beyond the scope of this paper to outline the possible range and quality of strength-based approaches [52], we also note the emphasis in Bryant et al.’s [52] work on foregrounding and embedding cultural practice as a way of ‘doing’ research (as opposed to being solely a factor of, or intervention in, the research process). Importantly, we note the alignment of this with the concepts inherent to salutogenic, complex systems approaches, particularly interconnectedness and emergence, articulated clearly in the following quote:

“*sociocultural approaches offer insight into the social mechanisms through which strengths and resources are ‘made’, and thereby offer a way to understand how these can be supported through programmes and other forms of social action*”. [52] (p. 1414)

### 3.2. A Positive Psychology Framework for Aboriginal Health Research: Tensions and Opportunities

Given this context, we propose two ideas. One, it is possible to see Aboriginal people be well, do well, and be very well. Two, adopting a positive psychology framework for understanding Aboriginal peoples’ health and wellbeing has relevance and utility in both conceptualising research and also in framing the research process in a practical sense. We argue that systems-informed positive psychology, as an extensible, flexible and strength-based framework, offers opportunity to understand, implement, interpret and promote what wellbeing means and can be for Aboriginal Australians. It does so in three clear ways: the theoretical mechanisms of positive psychology that influence wellbeing, the interactions and intersections within each of these factors, and the broader social context to enable enhanced wellbeing. However, it is important to acknowledge that there are areas requiring careful consideration throughout. We aim to outline a few important tensions inherent to positive psychology and opportunities that it is uniquely positioned to generate, address and/or further develop. While likely not a comprehensive list, we address some of the most immediate and salient points.

#### 3.2.1. Tensions

***Metatheoretical assumptions.*** The first is the question of relevance of the framework to Aboriginal people, particularly one that has been derived and sustained predominantly by non-Aboriginal practitioners up until this point [64]. Critique has been levelled at the limitations of positive psychology in the context of non-Western cultures, particularly around the very real impacts and consequences of oppression and social marginalisation, but also around the potential for poor fit to non-Western values and beliefs [65]. This is likely relevant in the Australian Aboriginal context also, though does not necessarily consider that Aboriginal peoples are highly diverse in our beliefs, values and indeed ideas of what it means to be Aboriginal [12]. Indeed, we argue that there are considerable human similarities shared by Aboriginal and non-Aboriginal populations alike, and much can be accomplished through a sharing of ideas, knowledge and models about the world and our worldviews within it, as noted by other Aboriginal and non-Aboriginal authors [11,64].

Importantly, recent research is beginning to explore the potential of positive psychology in global non-Western populations [66,67] and a small number of (non-Australian) First Nations populations [20,21,22,23]. Each notes the importance of the strength-based focus and its potential relevance and efficacy, while noting the necessity of understanding and deeply considering the cultural fabric of each population throughout the research process. In line with this, and the principles of true strength-based principles, it is vital that Aboriginal populations themselves determine and govern how these frameworks are adapted and implemented [36,67]. Importantly, the act of self-determining, adapting and implementing concepts of wellbeing and flourishing is influential beyond simply enacting voice and agency around one’s own circumstances. The extant literature in the First Nations context characterizes wellbeing as a socially constructed, oriented and defined concept that inherently rests upon relationality to culture, community, country and ancestors [14,20]. Indeed, the process, practice and values inherent to the act are deeply intertwined with a community’s identity, and thus serves as a mechanism to formulate and reformulate who and what the community is and what it represents, to those both within and outside of the community [49].

Interestingly, and also speaking to criticism of extant research approaches [52], it is the system focus of systems-informed positive psychology and its inherent foundation of ‘relationality’ that aligns strongly with strength-based approaches and facilitates an emergence of Aboriginal-defined wellbeing and flourishing through identity, practice and relationship. Notably, the act of self-determining wellbeing and flourishing itself contributes to wellbeing, and it is necessarily difficult to distinguish between individual and family or community wellbeing—they are deeply interconnected [14,16,20]. Indeed, it is in the central notion of interconnectedness across complex ecological and sociocultural systems that lies the tremendous and far-reaching potential for such a framework in terms of moving from an anthropocentric locus and conceptualisation of wellbeing to one where the catalysts of wellbeing are more difficult to pin down. Despite this, the ripple of wellbeing stemming from dynamic interactions between systems—including social, natural, artificial—is instinctively understood and experienced by all [36].

Finally, while this paper is about Aboriginal wellbeing, it is acknowledged that many non-Aboriginal people are engaged in this research context, and experience challenges in working appropriately and effectively with Aboriginal people [68,69]. This framework may also offer a means of better engaging with and understanding Aboriginal perspectives for those who do not walk in our shoes so that they too may enact and promote strength-based research that facilitates Aboriginal wellbeing and flourishing. Conversely, there are likely benefits for non-Aboriginal populations in utilising this framework for their own wellbeing—something of a ‘best of both worlds’ construct. The intersection of diverse knowledge systems has the potential to generate solutions and outcomes difficult or impossible to attain in isolation [70]. Inherent to this is a shift in the power dynamic at the epistemological level, but equally importantly at the sociocultural [2,70,71]. In this scenario, non-Aboriginal people cease to be solely the benefactor (that is, solving Aboriginal ‘problems’) and are able, or required, to assume the position of beneficiary (utilising Aboriginal knowledge to solve their own problems) [70]. This last point is becoming more relevant as we begin to understand the interconnectedness of our planet, our societies, and our knowledge systems and the utility of epistemic pluralism in approaches to solving many contemporary complex issues [72,73].

***False dichotomy: positive/negative: individual/system.*** Another tension we seek to address is the concept of dichotomous thinking and, within this, two meaningful and contested ideas. The first is inherent to positive psychology itself, and the perceived relentless focus on the positive [65], potentially ignoring the lived reality of many Aboriginal peoples. However, the field of positive psychology has matured over the past two decades by recognising negative states as normal, exploring their role in positive outcomes, and acknowledging the possibility of negative states as an outcome in itself [45]. The holistic nature of Aboriginal experience is fundamental to ideas of existence and of wellbeing [14,15,16], and the development and use of models should reflect both this notion of holism and also the possibility of variability within. Extending this, it is vital to frame this paper’s proposed approach as one not ignorant of the potential for, and deep importance of negative emotions and states [36,45]—‘normal’ states that often underpin and generate motivation and meaning for individuals and groups. Rather, the approach seeks to embody and support an agentic foundation. Motivation and meaning can be derived in many ways, for example, through the challenges, anger and rage borne of historical and contemporary issues, as noted elsewhere [65]. Beyond this, there is evidence that positive and negative cognitive states can coexist and that positive states may act as a buffer against mental and physical illness [74].

The second is that foregrounding system culpability alone ignores the fundamental inclusion of Aboriginal people within the system, and thus may have the potentially unanticipated impact of the erasure or suppression of individual agency. Argument has been, and should be, made that the focus on (for example) individual resilience may form a new ‘tyranny’ [52,65]. We suggest it is the conceptualisation of the fundamental idea of this that must be expanded, solely from that of individuals who are resilient to the harms of a system, to systems that facilitate and support resilience and growth [75]. While cognisant of the insidious nature of victim-blaming [11] and of the dire need for systems change and/or reformation to accommodate Aboriginal populations, we argue that there is value and importance in deep consideration of the potential hidden implications within the push for systemic change [76]. Simply, agency is imperative as it enables and propels communities to work from this foundation toward whatever end is considered relevant and meaningful for the individual and group [48,64]. Perhaps most importantly, Aboriginal people exist within, and are part of, a broader system, and the imperative and development of agency—as more than passive entities buffeted by forces outside of our control—is as vital to desired wellbeing outcomes as our efforts toward systemic change [48]. It is likely only through this model that we then can move to a further expansion of the earlier example of resilience—that of the establishment of systems that facilitate and support the dismantling of highly resilient problems [59].

Given this, we emphasise the utility of pluralism, an openness to multiple ways of seeing, understanding, improving, and solving challenges faced [77]. We suggest that the world is too varied, too changed for simplistic dichotomous thinking to further the wellbeing of populations. There is value in the exploration of possibilities for better outcomes and, importantly, how these are achieved, to whom flow the benefits, what these benefits are, and how they are sustained.

***The pre-eminence of culture.*** The importance of culture to the wellbeing of Aboriginal Australians is clear [78]. As primarily Aboriginal authors across this paper, we acknowledge culture as a genuine demonstrable determinant of wellbeing (see [61] for exceptions to this). However, while a core tenet of Aboriginal wellbeing and Aboriginal wellbeing research, we suggest that it would be potentially inaccurate, reductionist and constraining to state that culture alone determines what is meaningful for an Aboriginal person [79]. Indeed, this runs the risk of homogenising the phenomenological experiences of ‘being’ an Aboriginal person and the idea of ‘culture’ itself, for Aboriginal and non-Aboriginal populations alike [80]. Further, there may be unintentional harms associated with this for those who have experienced significant disconnection from culture, such as those of the Stolen Generations and their descendants [7]. Instead, we make a case for a multifaceted, strength-based complex systems conceptualisation to frame our health and wellbeing research activities. Such an approach can help to understand those broad strength-based ecological factors that support social and emotional wellbeing, physical wellbeing, and enable healthy long lives of a quality deemed relevant to and by Aboriginal peoples themselves.

Much of the available literature centers culture as a powerful protective factor for Aboriginal people’s health globally [81,82]. What remains unclear are the immediate impacts of culture broadly, and the likely inherent factors associated with culture and cultural practice (for example, being and belonging, concepts of agency, place and meaning). Likewise unclear are the broader systemic factors that interact and emerge, mediate, moderate and predict wellbeing or trigger restorative biological processes, beyond culture alone. We note the focus on the influence of culture in its own right [14,16,78], and as one component of culturally informed, strength-based interventions, but also as a means of understanding the mechanisms that underpin associations between culture and wellbeing, particularly physical wellbeing. Importantly, by framing this within systems-informed positive psychology, we enable the strength of culture to be understood more deeply, while enabling an understanding of the broader ‘ecology’ of Aboriginal Australians, the interactions and emergent properties across this ecology and how such things occur and can be leveraged toward Aboriginal wellbeing outcomes.

#### 3.2.2. Opportunities

***Deficit narrative implications.*** The first potential application of this model is simply to add, and offer new possibilities, to the body of work that aims to address pervasive deficit narratives of Aboriginal peoples. For far too long, Aboriginal health and wellbeing research have played a role in propagating ideas about Aboriginal people that are harmful [11]. This manifests in harmful discourse, perspectives, and understandings of Aboriginal populations, but also in the ways that research is conceptualised, implemented, evaluated and translated into policy and practice [83], thus further embedding and reinforcing problematised ideals. Both of these have implications for Aboriginal populations themselves, from the internalising of these deficit narratives, to the avoidance of, or walking away from, potentially helpful research or practices [11]. Each potentially leads to the problematic assumption that Aboriginal people do not want to, or simply cannot, help themselves—that we are in fact the problem [11].

A primary purpose of proposing positive psychology as a useful framework is thus to more fully realise a strength-based foundation from which to explore, bring together and better understand how to foreground the strengths of Aboriginal communities. Clearly this work is already underway [14]. Positive psychology’s focus facilitates a foundational foregrounding of those factors that promote salutogenesis, as opposed to shaping research through the lens of describing and ameliorating disease, risk factors and deficit [71]. The intention is not to ignore the very real health concerns of Aboriginal communities, nor to supplant the approaches, methodologies and philosophies noted earlier, but to add to, support, complement and extend their influence and impact [74]. A clear strength lies in the narrative that it both derives from and promotes—that is, that Aboriginal people are capable of wellness and thriving. However, another strength lies in its extensibility, in terms of facilitating greater understanding of existing theoretical approaches and their relevance to Aboriginal wellbeing, while also explicating and generating new models of efficacy and effectiveness.

***Metatheoretical potential.*** Following from the last point is the framework’s utility with much of its flexibility lying in its potential to act as connective tissue, accommodating a breadth of theoretical frameworks within. Perhaps most importantly, it facilitates orientation of the user toward salutogenic framings [50] and thus explication of existing theories, such as those underpinning agency [84], resilience [85], growth [25], optimism [86], and motivation [87]. This is important when considered in relation to research focused on understanding the intersection of Aboriginal and non-Aboriginal frameworks, specifically in the context of those things that lead to wellbeing and flourishing. The earlier example (Section 3.1 of this manuscript) focused on Aboriginal language revitalisation [56], when interpreted through (for example) Seligman’s PERMA model of wellbeing [88], makes clear this capability.

Moreover, while the example illustrates the ways in which research might align with suggested building blocks of wellbeing [88], it also implies how we might apply these ideas of salutogenesis to the conceptualisation phase of strength-based research, enabling us to think more clearly and foundationally about how to avoid deficit approaches [52]. For example, a research project may incorporate a co-design model to explore flourishing as a function of restorative practice on country, itself a function of the role of agency, relationality, positive emotion, or meaning making. Each of these then aligns with the fundamental tenets of positive psychology, and of culture and community strengths more broadly, and allows us to understand and describe factors underpinning human behaviour that are invoked or correlated with cultural factors that facilitate wellbeing.

It is within this metatheoretical capacity that the framework’s potential value becomes evident, bridging gaps and explicating connectedness between domains such as (for example) people and country. Simply adapting an intervention from one context for use in another is often fraught and limited in effectiveness due to its creation within a specific worldview [82]. Given this, we make the case that it is not ‘what you do, but how you do it’, if you are to replicate a given course of action toward wellbeing. Given the diversity of Aboriginal cultural groups across Australia, a means of understanding how these components of wellbeing link together to effect meaningful shifts in wellbeing is perhaps necessary at a level abstracted from ‘culture’. This allows us to develop interventions that are extensible and versatile, and have the potential to be co-configured by and with diverse Aboriginal groups to meet their needs while retaining the underlying intention of a move toward broad or specific elements of wellbeing and flourishing [53].

***Linking mental and physical wellbeing.*** A growing body of evidence points to the contribution of psychological functioning to improvement or maintenance of physiological functioning [51], including cardiovascular disease, diabetes, pain and all-cause mortality [89]. This is important given the noted chronic individual and systemic life stressors experienced by a disproportionate number of Aboriginal Australians and the contribution of these to poor psychological wellbeing and associated physiological markers of health [90]. It is well documented that wellbeing for Aboriginal people is holistic, beyond simply physical or psychological wellbeing, and emerges from a balance of many factors, including community, culture and country [14,16,81]. For example, a few recent interventions in the Australian Aboriginal context have focused on the role of culture and cultural practice upon the social and emotional wellbeing of Aboriginal populations (e.g., [56,60]). However, while associated with constructs inherent to positive psychology (for example, relationality, meaning or engagement), there has been limited rigorous examination of the potential physiological impacts of these concepts for Aboriginal Australians [81].

Extending this link, longevity studies internationally have noted links between psychological wellbeing and physiological health, above and beyond mitigation of physiological disease [68]. To successfully age, a range of internal and external factors are implicated. Some are modifiable, for example, individual dispositions and social connectedness [91,92], but there are also broader factors such as our ecological environments [93]. While these are influential in a number of ways (for example, through the promotion of adaptive health behaviours and the direct moderation of stress and disease), the limited evidence in the Australian Indigenous context also emphasises the necessity of systemic change to support individuals [89]. Importantly, this approach has implications for Aboriginal populations across the spectrum, from our Elders to our younger generations. The latter is particularly salient given the trend within Aboriginal health research toward conceptualising and optimising early life interventions and the significant weighting toward our younger generations in national targets [94].

In summary, we propose positive psychology as an extensible and flexible paradigm for Aboriginal wellbeing research, one that enables and draws upon a potent combination of humanistic, salutogenic, strength-based, collective, cultural and systems-informed factors and approaches. Through this, it offers a unique means of understanding the theoretical mechanisms within the broader framework that act to influence wellbeing, and the interactions and intersections with the broader social context that enable enhanced wellbeing. Importantly, it offers the opportunity to understand, implement, interpret and promote what wellbeing means and can be for Aboriginal Australians.

## 4. Discussion

This paper has explored the possibilities for Aboriginal wellbeing research through the lens of positive psychology, highlighting both opportunities and tensions of the framework in relation to Aboriginal wellbeing research broadly. In doing so, we aim not to supplant existing work/knowledge in this space, but to foreground the possibilities of a framework that appears to have increasing relevance to Aboriginal research more broadly. Within this, we work on the basis that many of the necessary ideas for improving Aboriginal health outcomes and wellbeing already exist, and it is how we understand and piece together the extant web of knowledge—practical and theoretical—that likely matters most at this point.

For this to be achieved, we propose an overarching model that enables salutogenic and strength-based research aligned with and determined by the needs of individuals and communities, while acknowledging and attending to the broader system of factors that play a role in wellbeing. It may be that this last point is the key. Evidence is beginning to point toward the interconnectedness of a range of factors that facilitate wellbeing [14]. However, it has not yet been explicitly articulated that it is what emerges from these factors in concert—and the how and why of this emergence—that is paramount.

### The Imperative for a Strength-Based Complex Systems Approach

One purpose of any given system is to maintain some conceptualisation of homeostasis or balance, whether in the context of social, natural, or artificial systems. For all practical purposes, and like any system, what exists in the Aboriginal wellbeing context at present appears intended to perpetuate the status quo [11]. Unfortunately, this has consistently marginalised those peripheral to the heart of its purpose and ‘makeup’—Aboriginal ‘unwellness’ appears to be the primary emergent property of the existing complex system. Given this, we need to ask what the ‘system’ is for, and how might it be reconceptualised to a new fundamental purpose [71]. If we think of the broader systemic context we exist within as a biological organism, it becomes clear that we need to conceptualise an environment If we think of the broader systemic context we exist within as a biological organism, it becomes clear that we need to conceptualise an environment we ourselves are an integral, agentic part of, not vestigial, anomalous, or simply subject to. This must be an environment that Aboriginal people are able to contribute to and benefit from as fundamental and vital agents in the systemic imperative to maintain homeostasis—to co-create ‘we-being’ [45].

Aboriginal societies and cultures have an inherent, intuitive understanding of the broader systemic context—ecological, environmental, political, social—and have sought and found balance for millennia [73,95]. It is important that we leverage this to understand the bigger picture of what makes us well as Aboriginal people, and how we strategically position and leverage these things to contribute to the wellbeing of populations, Aboriginal and non-Aboriginal. At present, we are tinkering at the edges in many ways [71]. A fundamental shift in the paradigm we are enmeshed within is much more likely to effect profound change [46] (p. 162), and it is becoming apparent that this is necessary across many marginalised populations [96]. This work has already begun (for example, [11,14,71]), and our proposition complements and extends this body of work. We suggest this paper’s proposed framework also feeds into and creates opportunity for Aboriginal Australians to shape broader conceptualisations of, and interventions toward, wellbeing. This has potential application across multiple levels and contexts, for example at the individual and community level, but also at the more abstracted level of shaping, measuring and monitoring national wellbeing [97].

In the Aboriginal Australian context, a focus on the mechanisms that underpin associations between culture and wellbeing appear to be a promising start. There is tremendous scope to extend research and interventions, with the emphasis on heightened holistic wellbeing and how it is generated through relevant approaches such as those raised in this manuscript. For example, a given approach may enhance the sense of psychological wellbeing through meaningful engagement with, and connectedness to, country, culture and practice, in turn also influencing physical outcomes [51]. However, it is important to note that solely focusing on interventions that seek to explore, articulate and act upon associations between (for example) cultural practice and wellbeing may well still be of limited utility for many health contexts if only considered within the relative isolation of the interventions themselves. This is particularly so in terms of what has the capacity to generate such wellbeing and the sustainability of these outcomes [36,45].

Without defining and understanding the systemic bounds of the environment a given intervention exists within, it is difficult to know why an intervention is or is not useful, and where the most useful points of intervention and leverage may be to support and sustain individual and community wellbeing and flourishing. At present, little is understood about what these things are or might be. Suffice to say, the complex web of interactions and agents is poorly reflected in our measures of wellbeing. Considered from this perspective, a systems-level, more abstract example of the alignment of Aboriginal wellbeing with a positive psychology framework—and the notion of powerful leverage points [46]—is important.

Here we suggest an example born from very recent discussion around measures of national wellbeing. Within this, we aim to illustrate how systems-informed positive psychology might help usefully define and influence the bounds of the broader system in terms of multidimensional wellbeing derived from the interconnectedness of known wellbeing factors [97]. There is a growing and powerful sentiment that existing measures of wellbeing have become increasingly misaligned with the needs of a nation (and indeed the planet) and fail to capture what is necessary for sustainable development going forward [97,98]. Gross Domestic Product (GDP) is commonly used by governments and policy makers as a proxy for the broader ‘wellbeing’ of a nation and its citizens [97,99]. Indeed, growth in GDP has become synonymous with an increase in overall quality of life for societies. However, GDP is an economic tool intended to measure levels of economic growth, not societal wellbeing, with this misapplication of an economic construct to human and social wellbeing having ideological origins [99]. This existing model and measure both ignore the vast majority of non-market-based contributors to wellbeing (for example, psychological, physiological, and ecological health). Of equal concern is that it ignores the resulting gross inefficiency, waste, and sustained harms [97,99]. Certainly, it does not consider a range of the things that are meaningful and increasingly measurable for many citizens, especially Aboriginal populations [14], and has been criticized for overlooking cultural diversity and associated social and economic inequities [98,99].

Problematically, existing measures of wellbeing (such as GDP) further reinforce the idea that somehow Aboriginal people are deficient in the things that non-Aboriginal society deems relevant, while incentivising and rewarding the system that plays a powerful role in ensuring this deficit. A perverse example of this is the misalignment between production and growth as a proxy for wellbeing and Australia’s health system, a system that “*assesses as positive any increase in medical spending by the population, even if it is due to poor health, stress and the spread of preventable diseases*” [97]. Given the state of Aboriginal Australian health and the enormous expenditure to ameliorate disparity, this perverse illustration should make clear the problematic cycle many Aboriginal communities are caught within, at levels we are generally not positioned to influence [98]. It should also have clear implications for the reader in terms of the broader Aboriginal wellbeing context, not simply for health, but also for education, employment and so on. This function of current systems, the harmful impacts on Aboriginal populations, and the call to recalibrate and reconfigure extant approaches, have been noted elsewhere [71].

Positive psychology informed by systems theory enables a paradigmatic shift from deficit language, framing and conceptualisation of traditional models and measures associated with wellbeing. It facilitates a move toward revised strength-based, culturally aligned, systems-informed wellbeing models that can go some way to a rehumanising of Aboriginal populations—and, ironically, non-Aboriginal populations. It enables an exploration and understanding of, and support for, broader, more meaningful conceptualisations of wellbeing and flourishing that are fundamentally interconnected in orientation and that encapsulate and are derived from human, animal, eco- and social systems [36,45,73,96]. In short, it may enable us to understand and monitor the broader ‘system of strengths’ that facilitates and promotes Aboriginal wellbeing. It is, of course, how we might embed these possibilities into our institutions, communities and broader society—and the means of measuring them—that is vital to consider. However, the approach and its potential align with our intent to consider Aboriginal wellbeing through a broad strength-based, systems-informed lens.

Thus, positive psychology offers the opportunity for a fundamental reconsideration of Aboriginal health and wellbeing in terms of incentives and indicators, and what determines and generates value, how and for whom [32]. At the same time, from a theoretical perspective, it is a fundamental reconsideration of the notion of positive psychology, the creation—perhaps ‘return to’ is more accurate—of an ‘Aboriginal’ positive psychology. It enables a conceptual framing that considers the cultural and ethical norms and values associated with the many groups across the nation. Further, it considers the distinct ideas of, and means of attaining, meaning, relationality, happiness, balance and ultimately wellbeing, and what these things might mean from an Aboriginal perspective [13,36,37,45,99]. However, we caution that a simple mapping of positive psychology frameworks is likely insufficient, and that deeper work is likely required to establish the parameters of an authentic, strength-based, culturally aligned Aboriginal conceptualisation of positive psychology [48,64]. To do this effectively and appropriately, we require frameworks that inherently move researchers toward the conscious design, implementation, translation and evaluation of research initiatives in ways that promote wellbeing, through all stages of the lifecycle [11,71]. We suggest that a vital first step is to undertake significant research to better understand factors that support, drive and maintain those Aboriginal populations that are flourishing, and indeed to determine and define Aboriginal flourishing itself [13]. Through this, we may both highlight potential approaches to address the challenges faced by Aboriginal populations that remain vulnerable and struggle, but also explicate ways to sustain shifts in outcomes through meaningful impact at policy and funding levels [100].

We are arguing, therefore, that a systems-informed positive psychology may be useful as a research lens that enables us to consider how we do, can and should think about and frame the research we do and its potential impacts [11]. It enables us to think about wellbeing from its very foundations, to question the assumptions we have about wellbeing and the research process and the relationship between the two [53]. The broader body of Aboriginal health research is difficult to quantify in social and economic terms [101,102]. Nevertheless, despite huge investments of time and resources, to date, there has been an inability to facilitate outcomes leading to sustained wellbeing [10] and a certain myopia regarding the big picture of Aboriginal health and wellbeing. For example, on the one hand, the volume of research on Aboriginal resilience is encouraging and suggests—arguably valorises—the strength of Aboriginal people [61,62,63]. On the other hand, this reflects a systemic environment within which we are unendingly required to be resilient, with an enormous human and economic cost that will inevitably require payment [103]. We reiterate Karlin’s sentiment that the model should sharpen our perspective of the terrain before us, and consequently the questions we ask of it—it is not there to fit the data. The purpose of our proposed approach is to enable understanding of the possible parameters and bounds from a strength-based context, while taking into account the broader context of Aboriginal peoples’ lived experience, and to move toward wellbeing as defined by us. Through a systems-informed positive psychology approach, we anticipate greater understanding of the complex system of interactors and agents that each of us as Aboriginal people live within, and the emergent phenomena of this broader environment. We suggest the potential to uncover novel evidence of the multisystemic factors that influence movement toward and maintenance of flourishing, despite adversity—not simply those that contribute to states of poor wellbeing and health. From this vantage point, we will be more equipped to develop models of, and promote systems that support, Aboriginal flourishing and wellbeing. These models—inclusive of the vital points of adaptation and emergence, resilience and, more importantly, growth—can then be utilised for future research and interventions when working with Aboriginal families, and beyond.

## 5. Conclusions

Significant destabilizing and disruptive global events over the last few years (for example, the COVID-19 pandemic and climate crisis) have, perhaps irreversibly, brought to the forefront of entire populations questions of life meaning, connectedness with and to each other and the planet, and the necessity of hope and joy. Aboriginal populations have arguably been at the forefront of this questioning since colonisation and the harms done to the very source of life’s meaning for so many. There is a good deal we can learn and apply through the development of such frameworks that are associated with, and attend to, factors known to contribute to human flourishing.

We reiterate and acknowledge the tensions associated with promoting this more nuanced approach, that it may be seen as a means of ignoring the real disparities in health outcomes between Aboriginal and non-Aboriginal populations, but also the tensions around homogeneity of Aboriginal identity and/or the centrality of culture. However, the relevance and salience of these concepts and frameworks has never been greater, and models associated with the impact of broader elements of flourishing in Aboriginal Australian populations are yet to be developed. We reiterate the urgent need to push back against and/or expand upon the predominant focus on individual and community deficit and dysfunction within many disciplines. We also reiterate calls to investigate the potential benefits and utility of integrating Aboriginal ways of knowing, being and doing and Western positive psychology methodologies (i.e., strength-based social science). We suggest this offers a unique and potent means of reflecting on the current status quo and enables new perspectives to shift the dial on seemingly intractable problems in our society.

The response to the call for truly strength-based research paradigms is growing in the Aboriginal space. While this paper contributes to this, we do not consider our proposed approach to be the only way, or even the most useful or usable. Rather, we aim to contribute a strength-based approach to Aboriginal health and wellbeing research that enables flexibility and scope to walk within the world as it is while respecting and foregrounding Aboriginal concerns and interests. While this is an important goal in and of itself, we think the impacts extend beyond this. We hope that our paper has stimulated further thought around the ‘how’ of doing Aboriginal research, certain tensions inherent to this proposition, and what this means to practitioners who aim to undertake strength-based Aboriginal research. We also hope our paper shines a light on the potential impacts and directions that this framework may enable and facilitate, while stimulating thought about culturally appropriate ways to do so. In true strength-based fashion, we hope that our paper stimulates thought around a functional inversion of the idea of Closing the Gap—that is, bringing non-Aboriginal Australians closer to an understanding of what enables wellbeing and flourishing for Aboriginal Australians, and thus all Australians.

## Data Availability

Not applicable.

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
