# Peer review of "From Deficit to Strength-Based Aboriginal Health Research—Moving toward Flourishing"

_ijerph, 2023, doi:10.3390/ijerph20075395_

Round 1

Reviewer 1 Report

Thanks for this interesting and challenging paper on the wellbeing of Aboriginal Australians, by Aboriginal peers and colleagues. The content is excellent but I find the language quite heavy with some long and convoluted sentences. Even for a person with English as primary language I would like to see the work re-written to make it easier to read and understand. The work may be over-ambitious leading to incomplete explanation of many of the concepts.

I also found many statements presented without adequate evidence being provided for an academic publication. This evidence could be based on the authors' own experience or research, or through a citation. I have described only a few of these in detail below but there are others which should be more critically examined for the work to be rigorous.

As a non-Indigenous Australian, many of the terms and concepts in the article are familiar to me but I think that for an international audience more information and background are needed. For example the lead authors' nation (Noongar) is likely unfamiliar to most non-Australians and a brief explanation of who are the Noongar people would enhance the work. Also the meaning and importance of stolen generations requires particular explanation because of its importance to the overall subject of the paper.  I note the comment that one of the authors is the first generation to be raised by family, but this denies the thousands of generations of Aboriginal Australians raised by their family before European colonisation of Australia.

I find use of the term “Well-documented” (line 54) to replace a concise explanation assumes knowledge and access to documents that may not be available to an international audience. Alternatively, this could be omitted and the work more focussed on the need to move beyond the deficit narrative.

I suggest the question of who defines flourishing for a particular individual requires consideration. Is flourishing a characteristic only of an individual? Or would flourishing for Indigenous peoples be a characteristic of families or communities?

This question is partly answered in the section of Discussion headed "The imperative for a strength-based complex systems approach" (line 486) which refers to the broader systemic context. However if the concept of wellbeing and flourishing of communities rather than isolated individuals had been included in the initial research question then more research in this area may have been identified. 

I think there is a need for a reference or other evidence to support the statement line 497 “Aboriginal societies and cultures have an inherent, intuitive understanding of the broader ecology, and have sought and found balance for millennia.” Ecology in particular can refer to environmental, social, systems or political ecology.

“Positive psychology has generated meaningful outcomes for many populations, with an increasing focus upon and relevance to non-WEIRD (White, Educated, Industrialised, Rich and Democratic) populations” (line 512). This sentence requires reference or other evidence.

I cannot understand the meaning of this sentence: “However, it is important to note that solely focusing on interventions that seek to explore, articulate and act upon these links in relative isolation may well be of limited efficacy for many health contexts (beyond merely psychological and physiological, neurological and neurotrophic effects of the interventions themselves), both in terms of what has the capacity to generate such wellbeing, but also the sustainability of these outcomes.” This is one of many sentences that I think could be re-written to make the work more accessible, especially for people whose primary language is not English.

Line 510: "At individual and community level, … salutogenic approaches can be beneficial,..." cites Carr et al. However the review by Carr et al examined only the impact of positive psychology interventions on individual wellbeing. Additional references are needed to provide evidence of other salutogenic approaches, and their impact on community level wellbeing.

GDP by definition refers to “domestic” product, and is not a measure of "how well a society is doing" as claimed. GDP is an economic marker not a wellbeing marker as the authors suggest. The overall relevance of the paragraph beginning line 529 to the paper needs to be clarified or the paragraph could be removed.

Paragraph line 576 claims that it is difficult to quantify the broader body of Aboriginal health research: however there are regular published quantifications of Aboriginal health research in terms of numbers of papers as the next sentence states.

Reference to COVID in the conclusion needs greater explanation of COVID and exactly what the authors intend to explain from this reference in the main text, or remove this single mention of COVID.

Overall the concepts and thesis of this work are important and innovative. However the explanation and evidence, and language used to describe them need development.

References have many errors, for example:

Fforde reference has paragraph break at wrong point

Gwynne reference needs new line, Graham reference has wrongly placed hyperlink

Hone et al and Wright et al references appear incomplete

Inconsistency around use of hyperlinks; quotation marks (eg Tedeschi reference)

Marmot & Wilkinson and Meadows books should refer to specific chapter or pages

Author Response

Dear Reviewer,

Thank you very much for your encouraging words and excellent feedback for our paper “From deficit to strength-based Aboriginal health research – moving toward flourishing”. We appreciate being given the opportunity to submit a revised draft of the manuscript for publication in the International Journal of Environmental Research and Public Health.

We appreciate the time and effort that you as a reviewer have dedicated to providing feedback on our manuscript and are grateful for the insightful comments and valuable improvements we have been able to make to our paper. Thank you.

We have incorporated most of the suggestions made by the reviewers. Those changes are highlighted within the manuscript itself, and also within a Response to Reviewers document. Please see below our attached response to your feedback, in red, for a point-by-point response to the reviewers’ comments and concerns. All page numbers refer to the revised manuscript file.

Kind regards,

The authors

Reviewer 2 Report

It is a very nice reading to me. Thank you. I think it's a very timely paper in the context of Covid as countries across the world are facing challenges in providing health care services needed by their diverse populations. The flourishing of indigenous populations like the Aboriginals in Australia is very relevant. Deficit based health research has been dominating in the world and a strength based research aligned with SIPP has the potential to 'shit the dial'.

Happy to know this is a concept paper, mainly by Aboriginal authors (with collaboration with non-Aboriginal), which is great.

I wonder if there can be a section of "Methods and Results", which can come mainly from 1.4-1.7, if appropriate. The "Introduction" seems to have included too much of this manuscript. In addition, the 'Conclusion" should be shortened with the key points only.

Author Response

(The authors gave the same response as above.)

Reviewer 3 Report

Dear authors,

The relevance of your text is clear and adequate for the context of the special issue “Health and Wellness for Indigenous Peoples”.

While relevant suggestions for improvement can be found below:

-       - The text follows more on the category of “review” not “article”.

-        -Topic 1.4 “Pilot testing a SIPP model of wellbeing” should be reformulated. A pilot research is mentioned but the details (for example, methodology and results analysis) are not clear.

-        -Since I deemed this submission as a scientific work, it lacks more literature background in some parts to corroborate your decisions/reflections(for example from line 486-494).

-        -The discussion starts as a discussion however it then disperses, particularly after the title “The imperative for a strength-based complex systems approach”, where it looks like new information is given. The discussion should be a place to reflect upon your discoveries and what the literature already says.

-        -Also, in the submission guidelines for this special issue it is stated that “All papers submitted for consideration should include a paragraph in the Methods section briefly detailing: (a) the nature of the engagement, and the involvement and leadership of Indigenous people and communities within the project; (b) ethics and governance considerations in relation to Indigenous peoples; and (c) whose priorities are reflected in the work (Griffiths et al., 2022).” However, this is not clear in your paper.

-       - Since you propose a framework, a figure with the overall variables and -relationships would be important to improve the clarity of your idea.

-        -The paper sounds like a mix between a personal and a scientific approach to the topic. While understanding the significance of the personal side, since it is intended as a scientific paper, a scientific approach should be the focus. So I would suggest rearranging the writing, for example, line 411, 412.

Nevertheless,, the manuscript focused on an important topic for the indigenous population and shows innovation in the idea of applying a more positive perspective. However, improvements in the paper are needed, as stated above.

Author Response

(The authors gave the same response as above.)

Round 2

Reviewer 1 Report

Thank you for incorporating contributions from reviewers, I think the work is stronger and more rigorous, and ready for publication.

I think more work to shorten and sharpen sentences would ensure the integrity of the work and increase its readership, value, and influence. I see the references are much more correct, but I note two references by Fogarty et al (11 and 84) appear to be presented in different ways thought they are in the same series of work.

Author Response

Dear reviewer,

Thank you again for your feedback. It has strengthened our manuscript.

All requested changes have been made. We have aimed to simplify the paper for readability as sugggested, this time through splitting/shortening long and convoluted sentences. There are numerous modifications throughout the document, so these are noted in red throughout paper. There are too many to list in a table.

We have also amended the noted references 11 and 84 to reflect how they differ (one about deficit narratives broadly, the other specific to recreating these narratives in policy). We have also added an omitted reference in the list, for  citation [99]. It was missing in the earlier iteration.

kind regards,

The authors
